# Reproducibility study of "Fair attribute completion on graph with missing attributes"

## Abstract

Tackling unfairness is a challenging task with extensive difficulties in the context of graph learning models. One of the major issues is posed by the absence of node attributes, due to missing data or privacy concerns. A recent work by Guo et al. (2023) titled "Fair attribute completion on a graph with missing attributes", tackles this problem by introducing FairAC. The framework's main components adopt state-of-the-art approaches, including a sensitive discriminator and an attention mechanism to provide a solution to both the unfairness and attribute completion problem. Supported by an experimental analysis, FairAC claims to exhibit superior fairness performance while achieving similar node classification performance compared to other baseline methods. In our work, we try to reproduce the results provided by the authors along with validating their main claims. On top of that, this analysis highlights FairAC's ability to handle graphs with varying sparsity and fill missing attributes, even in cases of limited neighbouring data.

## 1 Introduction

Graph neural networks (GNNs) have been actively used in the bibliography (Scarselli et al., 2008; Wu et al., 2020; Jiang et al., 2019; Zhu et al., 2021; Chu et al., 2021) and have proven to reveal excellent results in classification tasks e.g. node classification or link prediction in social networks (Bhagat et al., 2011; Zhang & Chen, 2018). However, GNNs among other machine learning models are prone to exhibiting unfair classification with regard to sensitive attributes (Dwork et al., 2012). Tackling the problem of ensuring fairness in graph neural networks remains a non-trivial task. FairGNN (Dai & Wang, 2021) is a framework that minimizes unfairness while maintaining similar performance using a novel method called adversarial debiasing. This approach involves training a classifier as a constraint to filter out sensitive information from original data.

Current approaches in fair graph-learning assume a complete graph, lacking nodes with missing attributes. Since, GNNs do not work on nodes that have missing attributes. HGNN-AC, a framework proposed by Jin et al. (2021) presents a new approach to solve the missing attributes problem in heterogeneous GNNs. FairAC, the framework of the paper subject to this reproducibility study, broadens the scope of FairGNN by enabling node classification in graphs containing nodes with completely missing attributes (Guo et al., 2023). This paper aims to reproduce the claims made in the work of Guo et al. (2023). In summary, our contributions are:

1. [**Reproducibility study**] Reproduce the findings of the original paper, by using the documentated implementation details, baseline methods, and the provided code and datasets. Notably, we address the challenges faced in this process including: the absence of explicit training code, lack of adaptation of the baseline models (FairGNN & GCN) and insufficient documentation on hyperparameters.

2. [**Extended work**] **Improvement of the original code**: The original code doesn't contain the complete set of models, scripts and training functionality to generate the experimental results displayed in the original paper. We complement the existing open-source implementation with the

missing parts and provide the trained models along with an evaluation code, to facilitate the reproducibility process.

3. [**Extended work**] **Ablation study**: Sparsity varying graph evaluation. To evaluate the robustness of the fairness in FairAC vs. FairGNN, we increase the sparsity of graphs. The results indicate that the attention based attribute completion method performs better than completion by averaging the neighbouring nodes.

## 2 Scope of reproducibility

The work by (Guo et al., 2023) addresses the problem of establishing fairness in graph-based machine learning. Ensuring fairness in tasks e.g. node classification needs to be tackled with regards to two distinct sources of unfairness. Addressing the first issue, coined feature unfairness, involves minimizing the unfairness within node attributes. Social networks like Facebook are based on a network of profiles, that have different attributes. Excluding sensitive attributes (e.g. race) from nodes in the network doesn't guarantee fair treatment in node classification, unfairness can still arise from race-specific attributes (Ma et al., 2022). Secondly, topological unfairness is observed when neighbouring nodes in the network exhibit similar features (Chen et al., 2020) (Mehrabi et al., 2021). Graph preprocessing methods, such as Deepwalk, are used to extract topological embeddings that reflect can reflect the relationship between nodes' attributes and those of their neighbors (Perozzi et al., 2014). As a result, they are able to expose sensitive-attributes in the generated representations.

While the topic of tackling unfairness in graph-based machine learning, has been investigated extensively (Dai & Wang, 2021) (Rahman et al., 2019) (Bose & Hamilton, 2019) (Hardt et al., 2016), broader applicability can still be obtained. Many graphs contain nodes with missing attributes, e.g. new Instagram users might have incomplete profiles. This issue, still poses a challenge for many of the existing frameworks (Jin et al., 2021)(Chen et al., 2020). FairAC employs attention to complete the presence of missing attributes (Vaswani et al., 2017). Specifically, attention is combined with a sensitive discriminator to solve the joint problem of graph attribution completion and graph unfairness. Notably, the subject study claims to have achieved the following:

- **Claim 1:** The FairAC framework achieves comparable classification performance, compared to state-of-the-art methods for fair graph learning.

- **Claim 2:** The FairAC framework considerably increases the fairness performance, compared to state-of-the-art methods for fair graph learning.

- **Claim 3:** The FairAC framework addresses attribute completion in graphs with largely missing attributes in their nodes.

- **Claim 4:** The adversarial learning part in FairAC is crucial for removing sensitive information, mitigating feature and topological unfairness.

- **Claim 5:** The FairAC framework is a generic approach that can be used to generate fair embeddings for different homogeneous graphs, making it useful for several graph-based downstream tasks.

## 3 Methodology

The code, datasets, and experiments for training and evaluating the FairAC model are publicly available on their GitHub repository [1]. However, the provided repository doesn't serve as a complete implementation to reproduce all experimental evaluations from the original paper. Notably, the results depicted in the original paper compare FairAC to several baseline methods: FairGNN, GCN (Kipf & Welling, 2016), ALFR (Edwards & Storkey, 2015), ALFR-e, Debias (Zhang et al., 2018), Debias-e, FCGE (Bose & Hamilton, 2019), BaseAC (Guo et al., 2023). Of these baseline methods, the authors only evaluated FairGNN and GCN to produce

---

[1]Original FairAC repository: `https://github.com/donglgcn/FairAC`

the results. The rest of the models' evaluation originates from the work of Dai & Wang (2021), and have been provided with complete attribute graphs. To reproduce FairGNN and GCN, we had to consult the code from the original FairGNN paper [2] Dai & Wang (2021). In FairAC, graphs are augmented to split the nodes into $\mathcal{V}^-$ and $\mathcal{V}^+$, respectively representing the nodes with missing and complete attributes. This split is controlled by the $\boldsymbol{\alpha}$ ratio. However, multiple augmentations for FairGNN and GCN were needed to introduce the $\alpha$-hyperparameter. Since these baseline models don't support graphs with missing attributes, an average-attribute completion method is implemented. Along with these augmentations, the codebase was complemented with the functionality to generate models and evaluate them in separate steps.

## 3.1 Model descriptions

Throughout the experimental evaluation, there are a total of four models that were tested: GCN, FairGNN, BaseAC, and FairAC. This subsection addresses the high-level workings of the different models.

### 3.1.1 FairAC

FairAC is a framework that tackles fairness aware attribute completion for graph machine learning tasks. Formally the framework works in the following manner:

**Formal description** Let $\mathcal{G} = (\mathcal{V}, \mathcal{E}, \mathcal{X})$ denote an undirected graph $\mathcal{G}$ with $N$ nodes $v_i \in \mathcal{V}$ and corresponding attributes $\mathcal{X}$ and labels $\mathcal{Y}$, connected by edges $\mathcal{E}$. The attributes of the graph are preprocessed by the topological embedding method: Deepwalk. It is an innovative method for acquiring latent representations of vertices and their relations within a graph (Perozzi et al., 2014). Let $\mathcal{S} = \{s_1, s_2, ..., s_N\}$ represent the set of sensitive attributes with $s \in \{0, 1\}$. Note that in this FairAC framework, we consider a binary-classification task, so $y \in \{0, 1\}$. The missing attributes are represented by dividing the nodes of the graph into two parts: $\mathcal{V} = \{\mathcal{V}^-, \mathcal{V}^+\}$. Where $\mathcal{V}^+$ consists of the nodes that have complete attributes, and $\mathcal{V}^-$ consists of nodes with missing attributes. Furthermore, $\mathcal{V}^+$ is divided into two parts $\mathcal{V}^+ = \{\mathcal{V}_{keep}, \mathcal{V}_{drop}\}$. The $\boldsymbol{\alpha}$-parameter controls the attribute missing rate within the graph $\mathcal{G}$ by ensuring that:

$$\alpha = \frac{|\mathcal{V}^-|}{|\mathcal{V}|} = \frac{|\mathcal{V}_{drop}|}{|\mathcal{V}^+|} \tag{1}$$

**Generating fair embeddings** To tackle feature unfairness, an autoencoder composed of encoder $f_E$ and decoder $f_D$ creates embeddings $\mathcal{H}_i = f_E(\mathcal{X}_i)$. To ensure the fair generation of feature embeddings, a sensitive classifier $C_s$ is introduced. The goal of the sensitive classifier is to predict the sensitive attribute $\hat{s}_i = C_s(\mathcal{H}_i)$ for an embedding $\mathcal{H}_i$. The feature unfairness mitigation loss function can be summarized as follows:

$$\mathcal{L}_F = \mathcal{L}_{AE} - \beta \mathcal{L}_{C_s} \tag{2}$$

By minimizing the feature unfairness, we will need to minimize the loss of the autoencoder $\mathcal{L}_{AE}$ and simultaneously maximize the loss of the sensitive classifier $\mathcal{L}_{C_s}$.

**Attribute completion** An attention mechanism is employed to aggregate the features of the neighbours of a node with missing attributes. Let $(u, v)$ denote two nodes that are neighbours that have topological embeddings $(T_u, T_v)$, the weight of this pair is calculated using the attention $att_{u,v} = \sigma(T_u W T_v)$, with $W$ a trainable weight matrix. The weight $c_{u,v}$ is computed by calculating the softmax of these neighbours of $u$ $(N_u)$:

$$c_{u,v} = \frac{exp(att_{u,v})}{\sum_{s \in N_u} exp(att_{u,s})} \tag{3}$$

---

[2]FairGNN repository: `https://github.com/EnyanDai/FairGNN`

Then, the feature embedding $\hat{\mathcal{H}}_u$ is obtained by calculating the weighted aggregation with multi-head attention:

$$\hat{\mathcal{H}}_u = \frac{1}{K} \sum_{k=1}^{K} \sum_{s \in N_u} c_{u,s} \mathcal{H}_s \tag{4}$$

After, the attribute completion loss is calculated as follows:

$$\mathcal{L}_C = \frac{1}{|\mathcal{V}_{drop}|} \sum_{i \in \mathcal{V}_{drop}} \sqrt{(\hat{\mathcal{H}}_i - \mathcal{H}_i)^2} \tag{5}$$

To mitigate the topological unfairness that can arise from completing attributes, we use the same sensitive classifier $C_s$ to predict the sensitive attributes. Consequently, we want to maximize this loss $\mathcal{L}_{C_s}$ to minimize the overall loss of FairAC, which can be calculated by:

$$\mathcal{L} = \mathcal{L}_F + \mathcal{L}_C + \beta \mathcal{L}_T \tag{6}$$

### 3.1.2 BaseAC

BaseAC is a simplified version of FairAC. It contains the attribute completion module, but lacks the feature- and topoligical unfairness mitigation modules. The purpose of BaseAC is to show the contribution of adversarial learning in FairAC. Therefore training the BaseAC is achieved by setting the $\beta$-parameter of the FairAC model to 0. This disables the adversarial learning loss terms, thus the model will not counteract unfair embeddings.

### 3.1.3 GCN

The model for GCN was adapted from FairGNN. The GCN is the only model that lacks a fairness mitigation technique. Since a default GCN is unable to handle missing attributes, the graph was preprocessed by averaging the neighbour's attributes to fill the empty nodes. Additionally, this model was augmented with an $\alpha$ parameter, representing the attribute missing rate, which the provided code did not implement.

### 3.1.4 FairGNN

The FairGNN model was adapted from the FairGNN original implementation [3]. Similar to FairAC, it uses a sensitive discriminator which is used to create fair embeddings. Furthermore, the same preprocessing step is taken, which averages all neighboring nodes to fill the nodes in the graph with missing attributes.

### 3.2 Datasets

Three public graph datasets are used in the experiments: **NBA** (Dai & Wang, 2021), **Pokec-z**, and **Pokec-n** (Takac & Zabovsky, 2012). With the NBA dataset being the smallest, it contains data on 403 basketball players, with as sensitive attribute whether the player has a U.S. nationality (Dai & Wang, 2021). The target is to predict whether the salary of the player is over the median or not. Secondly, the Pokec dataset is composed of anonymized data of users, originating from a Slovakian social network. The suffix of the Pokec-n and Pokec-z datasets represents the region from where the users are from and is subsequently the sensitive attribute. The target of these datasets is to classify the working field. More details on the datasets can be consulted in the Appendix A section.

### 3.3 Hyperparameters

In replicating the experimental evaluation, we adhered to the hyperparameters explicitly stated in the original paper. The implementation details covered the $\beta$ parameter, the attribute missing rate $\alpha$, and several

---

[3]FairGNN repository: `https://github.com/EnyanDai/FairGNN`

standard hyperparameters that remained unaltered throughout all experiments (e.g. epochs, learning rate...). Additionally, for the accuracy and ROC thresholds, we referred to the existing scripts available in the GitHub repository. Furthermore, in the original study, the seeds (40, 41, and 42) were used in training the methods.

**Hyperparameters summary**:

- **alpha** : a weight parameter from the FairGNN implementation that controls the covariance constraint in the loss function (Dai & Wang, 2021).

- $\beta$: a weight parameter that controls the fairness in the general loss functions of FairAC and FairGNN.

- **acc**: represents the threshold accuracy needed to evaluate the model. This follows the widely used evaluation protocol in fair graph learning.

- **ROC**: represents the threshold ROC needed to evaluate the model.

- $\alpha$: feature drop rate. It controls the proportion of nodes in the graph that have completely missing attributes (equation: 1).

### 3.4 Experimental setup and code

This subsection discusses the complete reproducible implementation of the experimental results gathered in FairAC of the four models discussed in 3.1.

#### 3.4.1 Modifications to the code

Since the original FairAC repository lacked the implementation of the GCN and FairGNN models, these were adapted from the FairGNN repository, along with code to train and evaluate the models. Moreover, this code was also extended in order to take into account $\alpha$ (attribute missing rate). Furthermore, while the provided code could only run in an Ubuntu operating system, we implemented the following enhancements: modified variable precision to int64 to ensure compatibility with Windows systems and introduced a new environment that is compatible with both macOS and Windows operating systems.

#### 3.4.2 Experiments

For the training of each model, we set 3000 epochs. The hyperparameters used for Table 1 and Table 2 can be found in Tables 6 and 7 in the Appendix A section.

To evaluate the classification performance of the models, accuracy and AUC metrics are used. To evaluate the fairness performance of the different models, statistical parity (SP) (equation: 7) and equal opportunity (EO) (equation: 8) metrics are used. These metrics are computed in the following way:

- Statistical parity: the probability of classifying $\hat{y}$ as positive, is equal for both values of the sensitive attribute ($s \in \{0,1\}$ and $\hat{y} \in \{0,1\}$) (Dwork et al., 2012).

$$P(\hat{y}|s=0) = P(\hat{y}|s=1) \tag{7}$$

- Equal opportunity: the probability of classifying $\hat{y}$ as positive, given that label $y = 1$, is equal for both values of the sensitive attribute ($s \in \{0,1\}$ and $\hat{y} \in \{0,1\}$) (Hardt et al., 2016).

$$P(\hat{y}=1|y=1, s=0) = P(\hat{y}=1|y=1, s=1) \tag{8}$$

All the scripts to evaluate the models are provided in the code which can be found in the Github repository.[4]

---

[4]Github repository for reproducing results: `https://anonymous.4open.science/r/FACT-64F6/README.md`

### 3.5 Computational requirements

All experiments were conducted on a NVIDIA A100 GPU with 40 GB HBM2. The experiments involving the Pokec datasets were significantly more computationally expensive than those of the NBA dataset. Specifically, the Pokec datasets needed about 45 mins for the training of 3000 epochs to end, while NBA needed 1 min approximately. The use of GPU resources across all experiments resulted in a total expenditure of 30 GPU hours.

## 4 Results

In this section, we present the reproduced results. As indicated by the original paper, FairAC is expected to exhibit superior performance compared to other baseline methods, particularly in terms of fairness. In the following tables, we observe that while this behavior is confirmed in some cases, it is not consistently observed in others.

### 4.1 Results reproducing original paper

Table 1 presents the reproduced results for FairAC, FairGNN, GCN, and other baseline models with an attribute missing rate of 0.3. The results for the baseline models (ALFR, ALFR-e, Debias, Debias-e, FCGE) where obtained from Dai & Wang (2021). Optimal outcomes are highlighted in bold. Although the numerical values show slight differences compared to those reported in the original paper, they generally align with the trends specified in the original findings, particularly for the Pokec-z and Pokec-n datasets. In these datasets, FairAC outperforms in fairness metrics, specifically $\Delta SP$, $\Delta EO$, and $\Delta SP+\Delta EO$, surpassing the other models, while maintaining comparable classification performance. This supports the fulfillment of both Claim 1 and Claim 2. However, in the NBA dataset, the results deviate from those reported in the original paper. One contributing factor is a misselection of hyperparameters during the training process.

Figure 1: Accuracy and $\Delta SP + \Delta EO$ of FairAC when varying $\beta$ on Pokec-z dataset with $\alpha = 0.3$

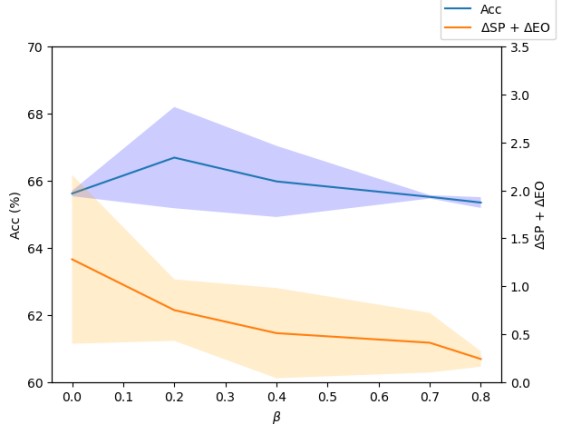

Table 2 showcases our attempt to replicate the original results, focusing on the comparison between FairAC and baseline methods across various attribute missing rates $\alpha$. Notably, FairAC exhibits superior performance in the fairness metric $\Delta SP+\Delta EO$, having the lowest values across all four levels of missing rates when compared to other models. Even in comparison to BaseAC, FairAC stands out due to its dedicated modules addressing feature unfairness and topological unfairness, thereby showcasing enhanced performance.

While some variations in metric values compared to the original paper are observed in both tables, these differences may be attributed to disparities in hyperparameter setups that are not explicitly stated in the original paper. Another potential factor contributing to the variations could be the split between keep and drop sets specifically, the selection of different nodes for these sets, which may impact the results. It is essential to note the sensitivity of the results to the selection of hyperparameters, particularly **acc** and **ROC**. Slight adjustments to these values can significantly impact the outcomes.

In Figure 1, we try to replicate the analysis across various values of the hyperparameter $\beta$. The accuracy metric values fall within the same range as those in the original figure. Interestingly, as $\beta$ increases, there is a slight decline in the accuracy of node classification. Conversely, the fairness metric values exhibit lower-than-expected results. Nonetheless, the observed trend aligns with the original paper, where the fairness metric decreases with an increase in $\beta$.

Table 1: Comparisons of FairAC and baselines for the studied datasets with $\alpha = 0.3$

| Dataset | Method | Acc ↑ | AUC ↑ | ΔSP ↓ | ΔEO ↓ | ΔSP + ΔEO ↓ |
|---|---|---|---|---|---|---|
| NBA | GCN | 70,58 ± 0,27 | 77,27 ± 0,17 | **1,03** ± 0,92 | **0,59** ± 0,76 | **1,61** ± 1,12 |
| | ALFR | 64,30 ± 1,30 | 71,50 ± 0,30 | 2,30 ± 0,90 | 3,20 ± 1,50 | 6,16 ± 3,10 |
| | ALFR-e | 66,00 ± 0,40 | 72,90 ± 1,00 | 4,70 ± 1,80 | 4,70 ± 1,70 | 9,4 ± 3,40 |
| | Debias | 63,10 ± 1,10 | 71,30 ± 0,70 | 2,50 ± 1,50 | 3,10 ± 1,90 | 5,6 ± 3,40 |
| | Debias-e | 65,60 ± 2,40 | 72,90 ± 1,20 | 5,30 ± 0,90 | 3,10 ± 1,30 | 8,4 ± 2,20 |
| | FCGE | 66,00 ± 1,50 | 73,60 ± 1,50 | 2,90 ± 1,00 | 3,00 ± 1,20 | 5,9 ± 2,20 |
| | FairGNN | **71,36** ± 0,81 | **78,60** ± 0,23 | 1,54 ± 0,71 | 3,59 ± 3,73 | 5,12 ± 4,41 |
| | FairAC | 70.73 ± 0.27 | 75.98 ± 2.30 | 1.22 ± 1.71 | 1.07 ± 1.56 | 2.29 ± 3.27 |
| Pokec-z | GCN | 65,74 ± 0,64 | 68,98 ± 0,22 | 1,38 ± 0,67 | 0,94 ± 0,83 | 2,32 ± 1,50 |
| | ALFR | 65,40 ± 0,30 | 71,30 ± 0,30 | 2,80 ± 0,50 | 1,10 ± 0,40 | 3,90 ± 0,90 |
| | ALFR-e | **68,00** ± 0,60 | 74,00 ± 0,70 | 5,80 ± 0,40 | 2,80 ± 0,80 | 8,6 ± 1,20 |
| | Debias | 65,20 ± 0,70 | 71,40 ± 0,60 | 1,90 ± 0,60 | 1,90 ± 0,40 | 3,8 ± 1,00 |
| | Debias-e | 67,50 ± 0,70 | **74,20** ± 0,70 | 4,70 ± 1,00 | 3,00 ± 1,40 | 7,7 ± 2,40 |
| | FCGE | 65,90 ± 0,20 | 71,00 ± 0,20 | 3,10 ± 0,50 | 1,70 ± 0,60 | 4,8 ± 1,10 |
| | FairGNN | 64,98 ± 0,90 | 68,42 ± 1,12 | 1,90 ± 2,12 | 2,35 ± 2,36 | 4,25 ± 4,48 |
| | FairAC | 65,29 ± 0,32 | 71,59 ± 0,26 | **0,25** ± 0,22 | **0,07** ± 0,04 | **0,32** ± 0,24 |
| Pokec-n | GCN | 68,83 ± 0,91 | **73,88** ± 0,71 | 3,11 ± 2,72 | 5,90 ± 1,01 | 9,02 ± 3,62 |
| | ALFR | 63,10 ± 0,60 | 67,70 ± 0,50 | 3,05 ± 0,50 | 3,90 ± 0,60 | 3,95 ± 1,10 |
| | ALFR-e | 66,20 ± 0,40 | 71,90 ± 1,00 | 4,10 ± 1,80 | 4,60 ± 1,70 | 8,7 ± 3,50 |
| | Debias | 62,60 ± 1,10 | 67,90 ± 0,70 | 2,40 ± 1,50 | 2,60 ± 1,90 | 5,0 ± 3,40 |
| | Debias-e | 65,60 ± 2,40 | 71,70 ± 1,20 | 3,60 ± 0,90 | 4,40 ± 1,30 | 8,0 ± 2,20 |
| | FCGE | 64,80 ± 1,50 | 69,50 ± 1,50 | 4,10 ± 1,00 | 5,50 ± 1,20 | 9,6 ± 2,20 |
| | FairGNN | **69,05** ± 0,62 | 71,18 ±0,54 | 0,76 ± 0,91 | 2,08 ± 1,41 | 2,83 ± 0,51 |
| | FairAC | 67,11 ± 0,75 | 72,27 ± 0,69 | **0,32** ± 0,20 | **0,60** ± 0,21 | **0,91** ± 0,12 |

Table 2: Comparisons of FairAC models with FairGNN and GCN for the studied datasets with four levels of attribute missing rates $\alpha$

| $\alpha$ | Method | Acc ↑ | AUC ↑ | ΔSP ↓ | ΔEO ↓ | ΔSP + ΔEO ↓ |
|---|---|---|---|---|---|---|
| 0.1 | GCN | 65.63 | 68.85 | 0.50 | 0.38 | 0.88 |
| | FairGNN | 65.26 | 68.78 | 1.09 | 1.09 | 2.19 |
| | BaseAC | **65.99** | 69.21 | 0.31 | **0.28** | 0.58 |
| | FairAC | **65.99** | **71.24** | **0.10** | 0.43 | **0.53** |
| 0.3 | GCN | **65.74** | 68.98 | 1,38 | 0.94 | 2.32 |
| | FairGNN | 64.98 | 68.42 | 1,90 | 2.35 | 4.25 |
| | BaseAC | 65.62 | 70.44 | 0.31 | 0.96 | 1.28 |
| | FairAC | 65.29 | **71.59** | **0.25** | **0.07** | **0,32** |
| 0.5 | GCN | **65.99** | 68.84 | 0.63 | 0.54 | 1.18 |
| | FairGNN | 65,65 | 68,92 | 0,83 | 0,81 | 1,64 |
| | BaseAC | 65.22 | 69.80 | **0.15** | 1.32 | 1.47 |
| | FairAC | 65.34 | **71.34** | 0.16 | **0.65** | **0.81** |
| 0.8 | GCN | 64.89 | 68.74 | 1.15 | 0.94 | 2.09 |
| | FairGNN | 65.54 | 68.79 | 1.14 | 0.91 | 2.06 |
| | BaseAC | 65.54 | 71.64 | 0.47 | 0.56 | 1.03 |
| | FairAC | **65.66** | **71.95** | **0.01** | **0.09** | **0.10** |

Table 3: Graph sparsity effect in missing attribute completion, using subsets of the NBA dataset with a feature drop rate equal to 0.5

| Median neighbor amount | Method | Acc | AUC | $\Delta$SP $\downarrow$ | $\Delta$EO $\downarrow$ | $\Delta$SP + $\Delta$EO $\downarrow$ |
|---|---|---|---|---|---|---|
| 68 | FairGNN | **79.23** | **80.37** | 2.34 | 0.0 | 2.34 |
|  | FairAC | 75.84 | 78.92 | **0.62** | 0.0 | **0.62** |
| 32 | FairGNN | 72.55 | **74.82** | 4.57 | 0.0 | 4.57 |
|  | FairAC | 72.55 | 73.04 | **0.65** | 0.0 | **0.65** |

## 4.2 Results beyond original paper

### 4.2.1 Missing attribute completion methods on sparse and dense graphs

One of the main claims of the paper is that it can deal better with nodes having missing attributes by utilizing an attention mechanism, to carefully interpolate information from neighbors. On the other hand, FairGNN (Dai & Wang, 2021) employs an average completion mechanism, when there are missing attributes. This series of experiments investigates further the association between graph density and the ability to fairly complete missing attributes.

In order to perform this experiment, we separate the NBA dataset into 2 train and test sets. Firstly, we calculate the median amount of neighbors for each node of the dataset. For a sparser setting, we will use as train and test sets the subset of the graph that has a number of neighbors lower than the median level. Accordingly, for the denser setting, we will use as train and test, the nodes with a neighbor amount over the median level. This will ensure for both experiments, that the information propagated between direct neighbors when filling missing attributes will vary.

The results presented in Table 3, show a clear superiority of FairAC for sparse graphs in terms of fairness since averaging between a few neighbors (FairGNN and GCN) can introduce a bias that the attention module of FairAC mitigates. Thus, as Dai & Wang (2021) argues, when having fewer neighbors a high bias from a neighbor can be transferred to the filled missing attributes when averaging; an issue that the proposed attention module can mitigate.

### 4.2.2 Generalization for other graph datasets

Training on different graph datasets provides the foundation to support further the claims of the original authors. It is necessary to use a graph dataset that incorporates sensitive information about the users. For this reason, we used the **German Credit Risk** dataset from Wang et al. (2022). This dataset can be originally found on Kaggle [5], containing information for users and their credit scores. Even though it is not originally a graph dataset, Wang et al. (2022) suggested a method to create edges between people based on credit similarity. In the context of our work, we used code from FairVGNN GitHub repository [6] and adapted it to our task.

Table 4: Comparison between methods for German Credit dataset, having as median 2 neighbors per node, and using 0.3 as the feature drop rate.

| Dataset | Method | Acc | AUC | $\Delta$SP $\downarrow$ | $\Delta$EO $\downarrow$ | $\Delta$SP + $\Delta$EO $\downarrow$ |
|---|---|---|---|---|---|---|
|  | GCN | 72.13 | **65.12** | 1.53 | 1.04 | 2.57 |
| German Credit | FairGNN | **72.40** | 63.44 | 1.98 | 0.76 | 2.74 |
|  | FairAC | 72.27 | 62.37 | **0.32** | **0.05** | **0.37** |

Table 4 shows the fairness overperformance of FairAC in comparison to other methods. German credit dataset is sparse, with a median amount of 2 neighbors per node. This means that the attention module

---

[5] German Credit Dataset Kaggle `https://www.kaggle.com/datasets/uciml/german-credit`
[6] FairVGNN Repository `https://github.com/YuWVandy/FairVGNN`

of FairAC can mitigate biases from neighboring nodes in comparison to FairGNN which averages all direct neighbors. Results from the German dataset, further support results from our previous experiment in Table 3, as well as the original authors' claims.

## 5 Discussion

Throughout this work, many experiments were made in order for results to be reproduced. Even though we had to extend the existing code and make several assumptions in terms of parameters, most of the results in the original work were feasible to reproduce.

In particular, the first claim states that FairAC achieves comparable classification performance with FairGNN and GCN. Our findings indeed show that FairAC doesn't affect classification performance in a major way. The second claim states that FairAC outperforms all baselines regarding fairness metrics. In our work, we verify that this holds true for both parity and equality metrics. The third claim, which states that the FairAC framework handles attribute completion in graphs with largely missing attributes in their nodes, was found to be supported since performance remained in good levels even when increasing the attribute missing rate. Furthermore, the fourth claim was validated by reproducing results and capturing the same impact of adversarial learning within the FairAC framework. This impact is noticeable by comparing FairAC with BaseAC. Regarding the fifth claim, the statement was validated to a large extent, by introducing the German Credit Risk dataset which is a sparse homogeneous graph of different types. It was noticed that FairAC performed well in this dataset both in terms of classification performance and fairness.

The original work was also extended, by providing insights regarding the impact of the FairAC framework when handling sparse graphs. It was noticeable that in such settings FairAC has superior performance compared to existing methods.

A major limitation of this work is that results are quite dependent on the selection of nodes with removed attributes. In this context, utilizing a pre-trained model could lead to different results in different test sets

### 5.1 What was easy

The original paper and repository provided essential information about the code for FairAC, making it straightforward to comprehend the fundamental concepts behind their work. Additionally, there was documentation on how to run the code, adding to its usability and accessibility.

### 5.2 What was difficult

The challenge arises from the absence of a proper code structure, posing a barrier to both the reproducibility and extension of the provided code. In particular, The paper didn't contain the necessary code neither hyperparameters for reproducing GCN and FairGNN models. The code was not clean and included many unnecessary classes that were not used throughout the paper. Additionally, some missing parameters required us to make assumptions for training the baseline methods. Finally, the incompatibility of the original environment with macOS and Windows posed another issue, contributing to difficulties in our work.

### 5.3 Communication with original authors

The authors promptly provided valuable feedback in response to our questions regarding the architecture of the models, hyperparameters, and implementation details.

### Broader Impact Statement

The user of this work should be aware that results are quite related to the selection of nodes of which we dropped attributes while training. In this context, it should be expected that a trained model could lead to different results for the same test set in case the attributes of different nodes are dropped.

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

## A   Appendix

### A.1   Datasets' statistics

In Table 5, we present the statistics of the three primary datasets used in reproducing the results of the original paper's experiments. Additionally, we provide the statistics of the new dataset, German Credit Risk.

### A.2   Experiments setup

In this section, we present the hyperparameter configurations utilized in our experiments to replicate the results of the original paper. The hyperparameters have been tailored to each dataset individually, as recommended by the original paper. In Table 6, we present the hyperparameters corresponding to the experiments outlined in Table 1. Similarly, in Table 7, we provide the hyperparameters for the experiments referenced in Table 2.

Table 5: Statistics of three graph datasets (Guo et al., 2023)

| Dataset | NBA | Pokec-z | Pokec-n | German Credit Risk |
|---|---|---|---|---|
| # nodes | 403 | 67,797 | 66,569 | 1,000 |
| # edges | 16,570 | 882,765 | 729,129 | 1,843 |
| Density | 0.10228 | 0.00019 | 0.00016 | 0.00184 |
| Median Neighbors per node | 47 | 9 | 9 | 2 |
| Sensitive attribute | U.S. nationality | region | region | gender |
| Target | median salary | working field | working field | good customer |

Table 6: Hyperparameters for Table 1 from original paper

| Dataset | Method | alpha | $\beta$ | acc | ROC | $\alpha$ |
|---|---|---|---|---|---|---|
| | GCN | - | - | 0.70 | 0.72 | 0.3 |
| NBA | FairGNN | 10 | 1 | 0.70 | 0.72 | 0.3 |
| | FairAC | - | 1 | 0.70 | 0.72 | 0.3 |
| | GCN | - | - | 0.65 | 0.69 | 0.3 |
| Pokec-z | FairGNN | 100 | 1 | 0.65 | 0.69 | 0.3 |
| | FairAC | - | 1 | 0.65 | 0.69 | 0.3 |
| | GCN | - | - | 0.66 | 0.69 | 0.3 |
| Pokec-n | FairGNN | 100 | 1 | 0.66 | 0.69 | 0.3 |
| | FairAC | - | 0.5 | 0.66 | 0.69 | 0.3 |
| | GCN | - | - | 0.71 | 0.61 | 0.3 |
| German Credit Risk | FairGNN | 10 | 1 | 0.71 | 0.61 | 0.3 |
| | FairAC | - | 1 | 0.71 | 0.61 | 0.3 |

Table 7: Hyperparameters for Table 2 from original paper

| Dataset | Method | alpha | $\beta$ | acc | ROC | $\alpha$ |
|---|---|---|---|---|---|---|
| | GCN | - | - | 0.65 | 0.69 | {0.1, 0.3, 0.5, 0.8} |
| Pokec-z | FairGNN | 100 | 1 | 0.65 | 0.69 | {0.1, 0.3, 0.5, 0.8} |
| | BaseAC | - | 0 | 0.65 | 0.69 | {0.1, 0.3, 0.5, 0.8} |
| | FairAC | - | 1 | 0.65 | 0.69 | {0.1, 0.3, 0.5, 0.8} |

