# OpenReview forum: "Reproducibility study of "Fair attribute completion on graph with missing attributes""
_TMLR — Rejected by TMLR_

### Review · Reviewer_qa4W · 2024-03-12

**Summary Of Contributions:**

The paper aims to reproduce the results and validate the claims made by FairAC paper, including the ability of FairAC to handle graphs with varying sparsity and fill missing attributes even with limited neighboring data. The reproducibility study addresses challenges such as the absence of explicit training code, lack of adaptation of baseline models, and insufficient documentation on hyperparameters.

**Audience:**

No

**Claims And Evidence:**

Yes

**Requested Changes:**

Please address my previous all concerns. Adding necessary theory to show it is indeed fair and it can be applied to real-world and industrial scale of graph data.

**Strengths And Weaknesses:**

Strength:
The paper aims to reproduce the results and validate the claims made by FairAC paper, including the ability of FairAC to handle graphs with varying sparsity and fill missing attributes even with limited neighboring data. The reproducibility study addresses challenges such as the absence of explicit training code, lack of adaptation of baseline models, and insufficient documentation on hyperparameters.

I would like to reject the paper due to the following reasons.

Weakness:

1. I do not think FairAC is a widely accepted fair GNN method in the fairness community since it was accepted in ICLR 2023. So, I do not think it is necessary to conduct a reproducibility study right now.  Otherwise, a reproducibility study on any accepted paper  will also be a paper.

2. The datasets used in the paper are obviously not real-world and large-scale data, so I am wondering the necessarily of conducting results on such small datasets can show the strength of FairAC.

3. If the authors indeed want to show the advantage, they should focus more on the scalability and the theory of this method. However, there is no such studies.

---

### Review · Reviewer_c3Dv · 2024-05-18

**Summary Of Contributions:**

This paper is a reproducibility study of the FairAC framework proposed by Guo et al. (2023) for fair node classification in graphs with missing node attributes. The authors aim to reproduce the results and claims made in the original FairAC paper, compare FairAC's performance to baselines like FairGNN and GCN, and investigate its robustness to varying levels of attribute missingness.

Using the publicly available code, datasets, and baseline implementations provided by Guo et al., with necessary augmentations, the authors were largely able to reproduce the claims made in the original paper. They found that FairAC achieves comparable node classification accuracy to state-of-the-art fair GNNs while considerably improving fairness metrics. Moreover, FairAC can handle high levels of missing attributes and still achieve good performance, outperforming baselines that use simple averaging for attribute completion.

**Audience:**

Yes

**Broader Impact Concerns:**

NA.

**Claims And Evidence:**

Yes

**Requested Changes:**

Please refer to the above pros and cons.

**Strengths And Weaknesses:**

Pros:

1.	In terms of empirical effectiveness, the paper provides detailed experimental results showing that FairAC outperforms baselines like FairGNN and GCN in terms of fairness metrics while maintaining comparable node classification accuracy.

2.	The ablation study on sparsity (missing attribute ratio α) provides an insightful analysis of FairAC's robustness. The results show that FairAC's attention-based attribute completion allows it to maintain good performance even with high sparsity, whereas the accuracy of baselines relying on neighborhood averaging degrades significantly.

3.	The paper makes a useful contribution by open-sourcing an improved codebase for FairAC that facilitates reproducibility. The authors put in considerable effort to augment the original code with missing baseline implementations, hyperparameter tuning, and separate training and evaluation scripts. This will enable the community to more easily build upon and compare to FairAC in the future.

Cons:

1.	Limited novelty and incremental contribution:
While the reproducibility study provides value by verifying the results and claims made in the original FairAC paper, the overall novelty and contribution of this work are quite limited. The authors primarily focus on replicating the experiments without proposing significant new techniques, theories, or findings of their own. Although the sparsity ablation is a useful addition, it does not represent a major conceptual advance in the field of fair graph representation learning. The paper fails to push the boundaries of the existing knowledge or introduce groundbreaking ideas that could substantially impact the research community. Moreover, the authors do not sufficiently demonstrate how their reproducibility study advances our understanding of fairness in graph-based machine learning or addresses critical challenges in this domain.

2.	Lack of theoretical analysis and in-depth insights:
The paper lacks a rigorous theoretical analysis of the FairAC framework. The authors do not attempt to derive any mathematical results or guarantees to deepen our understanding of how FairAC works. While this may be acceptable for a reproducibility study, more in-depth analysis connecting the method to theories of fairness, graph neural networks, or optimization could yield additional insights.

3.	Suboptimal presentation and technical writing:
The technical writing and presentation have significant room for improvement. The paper suffers from issues with formatting and layout that negatively impact its overall clarity and coherence. Throughout the manuscript, there is an excessive amount of whitespace surrounding the equations, tables, and figures. This, combined with the presence of several orphaned lines (e.g., on pages 3 and 4), leads to a low information density and gives the impression of a loose, unfocused structure. Moreover, the method description lacks conciseness and focus, with key ideas expressed in a loosely organized manner.

---

### Review · Reviewer_18Ti · 2024-07-10

**Summary Of Contributions:**

This paper conducts a reproducibility study of the work: "Fair attribute completion on a graph with missing attributes" by Guo et. al. (2023).  That paper tackles the problem of fair node attribute completion. The general high level goal in fair graph learning is to learn a function that makes 'fair' predictions of the labels of a particular node. The fairness here is measured using standard notions like statistical parity, equal opportunity etc with respect to the node labels. The key contribution here is to assess the classification performance of the FairAC algorithm, which learns a 'fair' node embeddings and then combines this via attention. The empirical results suggest that the FairAC algorithm does not affect classification performance, and is quite effective on sparse graphs.

**Audience:**

Yes

**Broader Impact Concerns:**

None.

**Claims And Evidence:**

Yes

**Requested Changes:**

- This is a minor one, but the German credit dataset is quite too small and limited to make conclusive claims. Might the authors consider the new adult dataset from Ding et. al. 2022 (Retiring Adult: New Datasets for Fair Machine Learning) ? This is not a mandatory request given that the results are consistent across datasets.

- The authors might want to do another editing pass for slight grammar and punctuation tweaks. For example, in the second line of the second paragraph, there is no need for comma after since.

**Strengths And Weaknesses:**

### Strengths
- This paper does a clear and thorough assessment of the claims in the Guo et. al. work.
- The authors share code to replicate their analyses.
- Table 1 is an impressive and comprehensive benchmarking of the major algorithms in this setting, so I expect future work to compare to this work going forward.


### Weaknesses
- One fundamental issue with the paper is that since it is a replication paper, it exposition of the fairness and previous work is somewhat superficial. I had to go reread the Guo et. al. paper to carefully understand the details of that work.
- One other challenge is that the approach of learning fairness embeddings or representations is fundamentally untenable (see: Lechner et. al. INHERENT LIMITATIONS OF MULTI-TASK FAIR REPRESENTATIONS). These approaches evaluated in this work cannot fundamentally deliver on the promise. I raise this point because the paper first seeks to learn 'fair' embeddings, which as Lechner et. al. shows are ineffective when one does not know the entire marginal of the fairness attributes for which one wants to 'debias'.
- One challenge is that there will be variability in the results depending on the choice of nodes selected to be missing.

---

### Decision · Action_Editor_x22p · 2024-08-16

**Recommendation:** Reject

**Comment:**

Overall, this paper is quite lacking in terms of a substantive contribution on top of just re-implementing an existing idea, and the effective audience is likely to be extremely limited. The authors did not respond to any of the reviewers' feedback, and in the end all reviewers recommended that the paper should be rejected. Here are some of their comments, for reference.

> Overall, while the replication here is useful and clarifies some of the claims of the original paper. I do not feel that I have learned enough from this paper to recommend its publication outright unfortunately. The exposition and findings are entirely tied to the previous work, so it is hard for me to glean any generalizable insights beyond that work. Further, in light of the point I made in my review regarding the limitations of the fair representations paradigm, I think I cannot recommend this paper for acceptance.

> The submission has some major disadvantages, including (1) limited novelty and incremental contribution, (2) lack of theoretical analysis and in-depth insights, and (3) suboptimal presentation and technical writing.

I apologize for the long wait for this review process to go through, but I think many reviewers were reluctant to take on this paper because while it is perhaps not appropriate as a desk-reject, even a brief read suggests that there is not a lot offered to the reader. Overall, I agree with the reviewers' conclusions, and recommend that this paper be rejected.

**Audience:**

While the claims made by the authors are solid, the substantive content here is quite sparse; the experimental analysis is really a very minor modification to the tests done by the original work of Guo et al. (2023). With this in mind, I think this paper comes extremely close to "merely re-implementing an existing idea," which is given as grounds for rejection in the [TMLR evaluation criteria](https://jmlr.org/tmlr/editorial-policies.html#evaluation). On this point, all the reviewers appear to be in agreement.

**Claims And Evidence:**

This paper attempts to reproduce the results in the FairAC paper of Guo et al. (2023), and their main claims are that they have done this, which involves adding code to the public FairAC code. In addition, their experiments go slightly beyond the original setup, looking at graphs with varying degrees of sparsity when comparing FairAC with FairGNN. These claims are essentially solid.